# Dry Needling in Physical Therapy Treatment of Chronic Neck Pain: Systematic Review

**DOI:** 10.3390/jcm11092370

**Published:** 2022-04-23

**Authors:** Manuel Rodríguez-Huguet, Maria Jesus Vinolo-Gil, Jorge Góngora-Rodríguez

**Affiliations:** 1Department of Nursing and Physiotherapy, University of Cádiz, 11009 Cádiz, Spain; mariajesus.vinolo@uca.es; 2Clinical Management Unit Rehabilitation, University Hospital of Puerta del Mar, 11009 Cádiz, Spain; 3Department of Physiotherapy, Osuna School University, University of Sevilla, 41640 Sevilla, Spain; jorgemgr@euosuna.org

**Keywords:** chronic pain, dry needling, neck pain, physical therapy

## Abstract

Chronic Neck Pain (CNP) is one of the main causes of disability worldwide, and it is necessary to promote new strategies of therapeutic approach in the treatment of chronic pain. Dry needling (DN) is defined as an invasive physiotherapy technique used in the treatment of neuromusculoskeletal disorders. The purpose of this review was to assess the effectiveness of invasive techniques in treatment of CNP. The search focused on randomized clinical trials, and according to the selection criteria, eight studies were obtained. In conclusion, DN can be an effective treatment option for CNP, positive outcomes were achieved in the short-term and in the follow-up performed between three and six months, and this technique may offer better outcomes than a placebo intervention based on the application of simulated DN.

## 1. Introduction

Cervical pain, or neck pain, can be defined as that unpleasant sensory and emotional experience associated with actual or potential tissue damage that affects the cervical region [1,2]. It may range from the suboccipital line to the level of the spine of the scapula [1,2]. Therefore, this condition is one of the main causes of disability worldwide, with a prevalence above 30% [1,3,4], which entails significant socioeconomic costs [1,4,5,6,7]. It becomes persistent in half of the cases, which exhibit chronic symptoms and recurrent pain episodes [3] that can extend beyond six months [5]. However, the updated classification of chronic pain allows us to understand chronic neck pain (CNP) as a primary entity that is not associated with a specific etiology, and lasts with functional limitation and emotional affectation for more than three months [8].

Studies indicate a female predominance in terms of the distribution by sex of neck pain, and in the age range of 35–49 years [9], especially from the age of 45 [10]. Typically, research indicates that the risk of neck pain is linked to physical and psychosocial factors, and may be related to lack of movement, sustained postures, and office work [11,12]. 

Usually, neck pain is nonspecific. This way, it is not attributable to fractures, trauma, or any other specific recognizable pathology (such as infectious, vascular, or oncological conditions). Therefore, examination and clinical analysis can rule out the warning signs that may relate the cases to specific systemic origins [1,3,5,13]. The assessment of patients with neck pain involves determining: (a) pain intensity by means of pain assessment scales (VAS or NPRS); (b) associated functionality or disabilities (Neck Disability Index, NDI) [14]; and (c) mobility of the cervical region (Range of Motion, ROM) [5]. Furthermore, in the complete evaluation of the neck, it is convenient to attend to the neurological assessment based on myotomes, dermatomes, and reflexes [15,16,17]. 

In addition, the assessment of patients with CNP should necessarily objectify comorbidities and associated symptoms [18], such as anxiety, depression, stress (DASS Scale) [19,20,21,22], and sleep disorders (Pittsburgh Sleep Quality Index) [20]. At present, it is essential to deepen the investigation of new strategies of therapeutic approach in the treatment of chronic pain, especially motivated by the low efficacy of the available pharmacological treatments. Therefore, it becomes convenient to look for alternatives that are effective and tolerable for patients [7,13].

In regard to physical therapy in the management of neck pain, the effect of conventional treatments is limited. Electrotherapy modalities (transcutaneous electrical nerve stimulation) could improve symptoms in CNP, but the evidence in this regard is not conclusive [23], and passive mobilization or manipulative therapy is no better than an exercise program [24]. 

Dry needling (DN) is defined as a minimally invasive physiotherapy technique used in the treatment of neuromusculoskeletal disorders [25,26,27]. Needling the most painful point of the muscle is also contemplated in traditional Chinese medicine acupuncture, where it is described as Ah Shi needling [28,29]. Its goal is to restore the physiological state of the tissue, reduce pain levels, and increase mobility through the application of mechanical stimuli caused by the insertion of acupuncture needles. These techniques are typical of physiotherapy, in which the physical agents pass through patient’s skin [25,26,27]. With respect to the classification of the needling technique, the purpose of classifying it as “dry” is to emphasize the condition of the physical agent, i.e., in this type of technique, there is neither pharmacological substances nor chemical agents introduced nor any fluid extracted [25,27,30].

Regarding the DN techniques, it is possible to define two modalities based on the depth of needle insertions [26,30]. The first is superficial DN, which confers analgesia by hyperstimulation. In this case, the needle goes through the skin and the subcutaneous cellular tissue without reaching the muscle. The other modality is deep DN, which functions directly on myofascial trigger points, since the needle penetrates the muscle tissue and has the ability to produce a local twitch response [26,30,31]. Local twitch response is an involuntary contraction reaction of the muscles to the mechanical stimulus of the puncture [31].

Thus, DN could be a treatment option for myofascial trigger points (hypersensitive areas of muscle fibers associated with motor abnormalities) [27]. However, precision during needling and the performance of the procedure seems to be essential for its correct development, with the ability of the physiotherapists being vital to perform the treatment properly [25,32]. The mechanism of action of DN is related to the effects achieved on myofascial trigger points [27]. The persistence of these points can favor the phenomenon of central sensitization. Therefore, it is possible to apply these invasive physiotherapy techniques in chronic pathologies [33], and it can be recommended for the treatment of CNP [27].

The present study arises from the need to deepen knowledge about the treatment of CNP through physiotherapy techniques. The goal was to assess the effectiveness of invasive techniques―specifically DN―in pain levels, and their relationship with other measurement variables, in order to establish action guidelines for the physiotherapeutic approach to CNP. Therefore, the main objective is to do a systematic evaluation of the effectiveness of dry needling in the treatment of chronic neck pain.

## 2. Materials and Methods

The present study is a systematic review addressing the topic being assessed in order to meet the effectiveness of dry needling in the treatment of chronic neck pain. The search focused on randomized clinical trials in order to obtain results that might indicate the most appropriate invasive physiotherapy intervention modalities in the treatment of CNP. The present work was conducted following the Preferred Reporting for Systematic Reviews and Meta-Analyses (PRISMA) guidelines, establishing the research approach through the PICO question format, namely: the selected population was the one that suffered from CNP (P = population); the intervention was invasive physical therapy treatment with DN (I = intervention) in comparison to other treatment modalities, or absence of treatment as control (C= comparison); and the main variable of the study was pain, which could be related to cervical mobility, quality of life, and other associated aspects (O = outcomes).

The bibliographic search was conducted in PubMed, Web of Science, Scopus, and Cochrane Library as reference databases within health sciences, and PEDro, as a specific database of evidence-based knowledge within physiotherapy. The search was conducted between October 2021 and March 2022. The descriptors “chronic neck pain” and “dry needling” were entered using the Boolean operator “AND” in databases. Therefore, the formula used was ““chronic neck pain” AND “dry needling””. The search was focused on these terms in order to analyze the updated evidence easily accessible to the health professional. It would be possible to include more combined terms; however, the intention was to show the results that the reader could quickly find. The research does not apply terms such as arthritis, fibromyalgia, or whiplash, since the search focuses on chronic neck pain and a primary origin not associated with trauma or systemic cause.

The following selection criteria were established so that the search was limited to clinical trials and prospective studies. This review includes only randomized clinical trials, and the randomization minimizes selection bias and favors similarity between groups [34,35]. 

Those studies included assessments of neck pain as one of the main study variables. Chronic pain is currently defined according to the new classification of World Health Organization [8], and the inclusion criteria establish the selection of patients with chronic neck pain exclusively. The selected studies start from subjects with chronic neck pain not associated with a traumatic origin. Interventions dedicated to physical therapy treatment were also inclusion criteria, and studies that focused on traditional Chinese medicine acupuncture were excluded. 

The studies discarded were those with repeated references, articles in languages other than Spanish or English, systematic reviews, study projects, case reports, studies of other pathologies or non-physiotherapeutic techniques or not conducted in humans, and those that were not considered relevant. The articles resulting from the search guidelines were analyzed in detail, and the selection of articles included investigations that had valid measurement instruments. 

The methodological quality of the studies was assessed through the score they achieved in the PEDro scale. PEDro is Physiotherapy Evidence Database, and this database is the main reference for finding out the most up-to-date evidence in physiotherapy, and for assessing the effects of interventions and treatments. The PEDro scale includes 11 items that allow assessing the methodological quality of randomized clinical trials with a final score range of 0 to 10; the items included in the PEDro scale can be seen in Appendix A. The PEDro scale has excellent reliability for use in systematic reviews of randomized clinical trials [36]. The scores were obtained from the PEDro database and later revised. 

Two independent reviewers performed the search and screened the articles; these reviewers applied inclusion/exclusion criteria, and later, another reviewer supervised the systematic review, quality assessment, and data extraction. The authors decided to make a qualitative analysis due to heterogeneity in outcome measurement precluding statistical integration with guarantees.

## 3. Results

According to the search and selection criteria previously established in the method of the present study, eleven studies are obtained. The PRISMA flow diagram (Figure 1) illustrates the conduction and selection stages of the systematic review.

Table 1 presents the characteristics of the clinical trials based on the application of DN in the treatment of CNP assessed (the sample was composed of 807 individuals with CNP). All included articles were randomized clinical trials. In addition, it indicates the methodological quality of these studies based on their score obtained in the PEDro scale. This scale assesses the level of recommendation of scientific articles based on their methodological quality, establishing a score between zero and ten. All the selected studies obtained a minimum score of five, and most of them reached a score of seven or eight points, which is a high recommendation level. This table also indicates the outcomes measures in the studies and the assessment time. PEDro score details of each randomized clinical trial selected are available in Appendix A. On the other hand, Appendix B includes specific details of each article, such as countries where studies were conducted, or the type of clinical center. Moreover, this table indicates the outcomes measurements and the assessment time of each randomized clinical trial selected.

The intensity of pain (VAS or NPRS scales) was the most assessed variable in the studies [29,37,38,39,40,41,42,43,44,45,46], followed by disability (NDI or NPQ scales) [37,38,39,40,41,42,43,44,45], and the pressure pain threshold (PPT) [37,39,41,42,45,46]. Five of the studies had assessed the cervical range of motion (ROM) [29,37,39,40,42], and other variables had also been included to a lesser extent, such as strength [39], perceived effects [43,44], self-efficacy [43], level of catastrophism [38,40,43], sleep quality [43], kinesiophobia [38,45], anxiety [38], depression [38], fear of pain [38], or attitude towards pain [38]. The length of follow-up varied between immediate post-intervention evaluation [29,46] and one year [44].

Table 2 shows the data referring to the intervention protocols (DN treatment and alternative treatment) of each of the clinical trials included in this systematic review, and the results of each study. 

The sample assessed was composed of 807 individuals with CNP, of which 398 had received physical therapy treatment with DN alone [37,41,46] or in combination with other complementary interventions [38,39,40,42,43,44,45]; 373 had received alternative treatment with different modalities based on manual therapy, such as stretching [37,38,39,40,42,43,44,46], therapeutic exercises [43,44], shock waves [41], kinesiotaping [40], transcutaneous electrical nerve stimulation (TENS) [38], microwave [38], or simulated DN [44,46]; and 36 received three treatments options (DN, needle acupuncture at distant point, and sham laser acupuncture with a 1 week wash-out period between the interventions) [29]. 

Treatments based on the isolated intervention of DN [41,46], including post-needling stretching [38,39,40], or combined with other therapies [38,42,43,44] were proposed in comparison to other treatment modalities, including placebo treatments using sham DN [29,42,46]. The intervention protocols ranged from a single treatment session [29,46] to a four-week treatment with up to seven sessions [44]. All the studies collected were randomized clinical trials, which entailed high methodological quality and in-depth analyses that allowed making comparisons and drawing significant conclusions.

The duration of the intervention in the trials was variable; likewise, the follow-up time indicates differences between investigations. Studies indicated positive effects on pain [37,38,39,40,41,42,43,46], NDI [38,39,40,41,42], ROM [37,39,40,42], PPT [37,39,41,42,46], strength [39], and other psychological factors such as kinesiophobia, catastrophic thinking, anxiety, depression, fear of pain, or attitude towards pain [38,40].

## 4. Discussion

The present review examined the most recent evidence available on the use and benefits of DN in physical therapy treatment for CNP. 

Pain intensity was the most studied variable. Depending on the study, the VAS scale or the NPRS scale were used, both of which showed high reproducibility and validity for short- and long-term assessments of CNP [39,41,45]. Focusing on pain, the shorter-term outcomes were found in the study conducted by Stieven et al. [46], who demonstrated the immediate effects of a single-session treatment. That study showed that a single application of unilateral DN at the level of the upper trapezius or a myofascial release treatment of that musculature could generate a superior response than a placebo intervention, with pain reduction and increased PPT.

Along the same lines, Sobhani et al. [40] performed a treatment of five sessions distributed over ten days, collecting the outcomes at the end of the intervention. These authors observed a decrease in the intensity of pain, a reduction in the NDI and catastrophic thoughts, and increased mobility. Disability is one of the important variables to assess in CNP, and usually the NDI scale is used, but it is also possible to use other scales [37]. Manafnezhad et al. [41] found similar effects in the follow-up performed one week after the intervention and after three weeks of treatment at the rate of one session per week.

On the other hand, it was possible to find the outcomes achieved by carrying out a long-term follow-up of up to one year (Gattie et al. [44]), also in comparison to a placebo-type sham DN treatment. These authors did not observe differences between DN treatment and placebo. Alternatives interventions based on placebo could suggest that the use of placebo could have a place within the treatments. In the same way, Irnich et al. [29] compared the effects of DN intervention versus traditional acupuncture treatment and sham laser treatment in the same group of patients. 

Most studies performed intermediate follow-ups ranging from three to six months [38,39,43], with four to six treatment sessions distributed over two to four weeks or one-month follow-up after two sessions with a one-week interval [42]. Regarding the periodicity of the follow-ups, it is worth highlighting the study conducted by Cerezo-Téllez et al. [39], whose analysis included up to six post-intervention assessments.

Upper trapezius and levator muscles are the most frequent locations to DN intervention [40,41,42,46]; usually, the treatment of studies includes DN in this musculature, and combine with other neck or back muscles [38,39,43,44]. Another aspect to analyze would be the performance of the technique unilaterally or bilaterally, although this would be related to the lateral predominance of the symptoms and to the proprioceptive control at the cervical level, as in cases in which there is a structural alteration [47].

In general, the applications of DN techniques were performed following the action protocols described by Travell and Simons [38,39,42], with rapid needle entry and exit movements under the principles of the Hong’s technique, in which the needle is retracted into the subcutaneous tissue and then redirected to another region of the trigger point without leaving the tissue [37,38,39] by means of the therapists’ wrist flexion and extension movements [41]. The procedure affected the musculature bilaterally [40], for one to two minutes [41], seeking to trigger local spasm reactions [38,39,42,43]. In many cases, DN was accompanied by ischemic compression or post-needling stretching [38,39,40,42,43,45].

The mechanism of action of DN can be determined based on chemical and neurophysiological changes associated with mechanical effects derived from the stimulus provided by invasive therapy on soft tissue [41], which modifies the activation and perpetuation of myofascial trigger points [42]; usually, the DN intervention causes a local twitch response [29,38,39,41,43,44]. The methodology proposed in the assessed studies focused on DN interventions on the myofascial trigger points of the upper trapezius and the levator scapulae muscles [40,41,42,46], and, to a lesser extent, on splenium, multifidus, or middle trapezius, among others [38,39,43,44].

In the studies that performed placebo interventions with sham DN [42,44,46], sham needles were used to simulate the puncture without penetrating the skin [42,44]. Therefore, three of the studies apply sham DN as a placebo treatment option, making it necessary to delve into the conditions of this intervention. In addition, the alternative treatment was performed by means of stretching [38,39], therapeutic physical exercises [43,44], or manual therapy techniques (myofascial treatment or cervical and thoracic mobilization) [40,42,43,44,46], or by means of instrumental techniques, such as TENS and microwaves [38], kinesiotaping [40], and waves shock [41]. 

The research of Leon-Hernandez et al. [45] stands out for the comparison between two treatment modalities based on the percutaneous needle electrical stimulation after application of the DN. In these treatments, the DN of the upper trapezius is performed (with local twitch response), and then a low or high frequency current is applied. This option shows that DN can be combined with associated electrotherapy and can obtain similar results regardless of the stimulation frequency.

Specifically, in the comparison between DN interventions and alternative treatments, it should be noted that the results may be favorable to invasive treatments [38,39,42]. However, the differences may be slight [43], or the beneficial effects achieved may be similar to those produced by the control treatment [40,41].

In general, it highlights the relationship of the treatment proposals of the trials with therapeutic exercise, and this reinforces the need to direct physiotherapy to a relationship between passive techniques and active movement. Exercise has positive effects on pain and functionality, and it should be oriented according to the interests and individual goals of the patient, and could be combined with instrumental techniques [48,49]. Is it possible to achieve the same effects with manual stimulation of the treatment points? [50]. 

The positive outcomes that support the success of DN with respect to the study variables in CNP are in line with the conclusions of other previous reviews that considered this type of intervention useful [51]. In addition, the changes achieved are in line with what has been observed in other related pathologies, such as headache [52,53]. In the same way, it would be possible to point out that these effects could help reduce over-medicalization, and represent a non-pharmacological treatment option [7], which will also reduce the socioeconomic costs associated with neck pain.

The results found show us that the research that relates the DN intervention with CNP is growing, with most of the available articles being recent. The limitations of the present review were due to the differences in the articles analyzed in terms of treatment protocols and lengths of follow-up. With a view to future clinical trials, it would be interesting to have tools that assess objective changes in muscle function or performance, with greater presence of strength tests and novel tools, such as electromyographic control. Consequently, it would also be convenient to study other aspects, such as the relationship between the results achieved with DN and the size of the needles or the duration of the session, and, above all, the type of employment of the patients, because the effect could be limited in work with static positions or head-down postures [11,48].

## 5. Conclusions

In conclusion, it is possible to point out that DN can be an effective treatment option for CNP. The studies assessed indicated that positive outcomes were achieved in the short-term and in the follow-up performed between three and six months, although the effects seemed to be limited in very long-term follow-ups, such as one year.

DN may offer better outcomes than a placebo intervention based on the application of simulated DN. This way, further research on this topic should be conducted. The recommended length of DN treatment for CNP would range from four to six sessions, distributed over two to four weeks.

The physiotherapy treatment based on the application of DN is mainly focused on performing the technique on the upper trapezius and the levator scapulae muscles following the procedures described by Travell and Simons. This intervention is normally performed bilaterally. It can be accompanied by stretching and combined with other techniques of manual therapy and therapeutic exercises. In addition to having effects on the intensity of CNP, DN treatments have had positive effects on other related variables such as ROM, NDI, or PPT.

Further studies are needed to combine the monitoring of short-and long-term variables, preferably in comparison to placebo interventions. Those studies will allow determining the changes induced at the structural and functional levels of the affected musculature, such as changes in the levels of strength or in the patterns of muscle activation derived from the interventions.

The variability among studies could make it difficult to determine conclusions. 

## Figures and Tables

**Figure 1 jcm-11-02370-f001:**
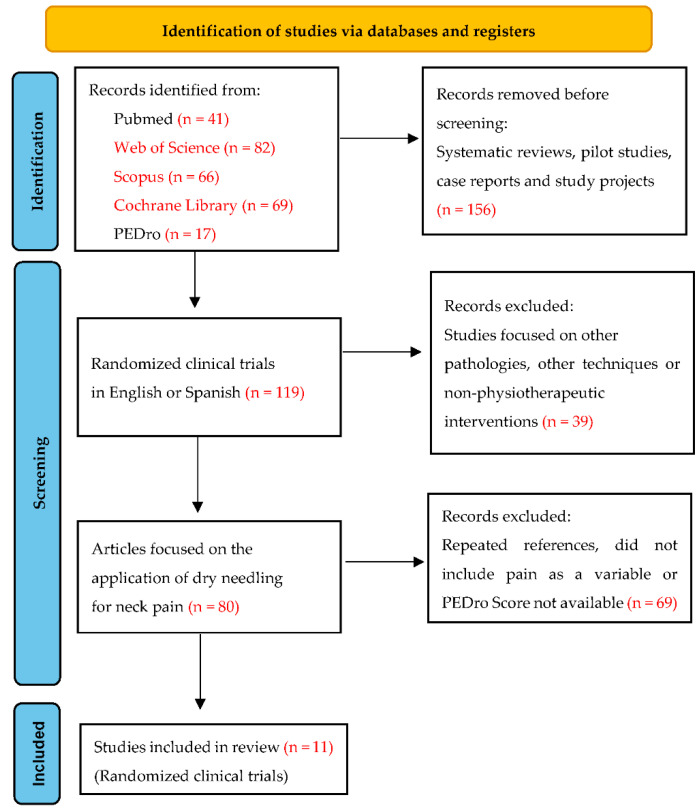
PRISMA flow diagram. Identification of the results obtained from the databases.

**Table 1 jcm-11-02370-t001:** Characteristics of the clinical trials included in the systematic review.

Author (Year)	Participants and Groups	PEDro Score	Outcomes Measurements	Assessment Time
Irnich et al. (2002)	N = 36	6/10	Pain ROM	Immediate post-intervention (15–30 min after treatment)
Llamas-Ramos et al. (2014)	N = 94 (47/47)	8/10	Pain PPT ROM Disability	3 post-intervention evaluations: 1 day, 1 week, and 2 weeks after the last treatment session
Cerezo-Téllez et al. (2016)	N = 130 (65/65)	6/10	Pain PPT ROM Strength NDI	6 post intervention evaluations: After 2 sessions; after full treatment; 15, 30, 90, and 180 days
Sobhani et al. (2017)	N = 39 (13/13/13)	5/10	Pain Catastrophism ROM NDI	1 post-intervention evaluation
Manafnezhad et al. (2019)	N = 70 (35/35)	6/10	Pain NDI PPT	Evaluation prior to each session and final evaluation 1 week after the last session
Gallego- Sendarrubias et al. (2020)	N = 101 (47/54)	7/10	Pain PPT ROM NDI	3 post-intervention evaluations: an evaluation after each session and one month after completion
Stieven et al. (2020)	N = 116 (58/58)	8/10	Pain NDI Perceived effects Catastrophism Sleep quality Self-efficacy	3 post-intervention evaluations: at 1, 3, and 6 months
Gattie et al. (2021)	N = 77 (37/40)	7/10	NDI Pain Perceived effects	3 post-intervention evaluations: 4 weeks, at 6 months, and 1 year
Leon-Hernandez et al. (2021)	N = 40 (20/20)	7/10	Pain PPT NDI Kinesiophobia	2 post-intervention evaluations: 1 week and 1 month
Stieven et al. (2021)	N = 44 (15/14/15)	8/10	PPT Pain	Immediate post-intervention evaluation and at 10 min
Valiente-Castrillo et al. (2021)	N = 60 (21/20/19)	8/10	Pain NDI Kinesiophobia Catastrophism Depression Anxiety Fear Pain Pain Attitudes	3 post-intervention evaluations: at the end of the full treatment, at 1 month, and at 3 months

Abbreviations. PPT: Pressure Pain Threshold; ROM: Range of Motion; NDI: Neck Disability Index.

**Table 2 jcm-11-02370-t002:** Interventions, procedures, and results of the clinical trials based on the application of DN in the treatment of CNP.

Author (Year)	DN Interventions Protocols	Alternative Treatment	Results
Irnich et al. (2002)	1 session DN trapezius, splenius, levator scapula, semispinalis, sternocleidomastoid, scalenus, and paravertebral muscles (LTR: Yes)	1 session: needle acupuncture at distant point/sham laser acupuncture	There are no differences between DN and sham laser acupuncture
Llamas-Ramos et al. (2014)	2 sessions in 2 weeks: DN upper trapezius (LTR: Yes)	2 sessions in 2 weeks: trigger point manual therapy (compression, stretching, and friction massage)	↓ Pain ↑ PPT ↑ ROM ↓ Disability
Cerezo-Téllez et al. (2016)	4 sessions in 2 weeks: DN multifidus, splenius, upper trapezius, and levator scapula (LTR: Yes) + passive stretching	4 sessions in 2 weeks: Passive stretching	↓ Pain ↑ PPT ↑ ROM ↑ Strength ↓ NDI
Sobhani et al. (2017)	5 sessions in 10 days: bilateral DN upper trapezius and levator scapulae (LTR: not specified) + passive stretching	5 sessions in 10 days: manual therapy (ischemic trigger point compression)/kinesiotaping on trigger points	↓ Pain ↓ Catastrophism ↑ ROM ↓ NDI
Manafnezhad et al. (2019)	3 sessions, 1 per week: DN upper trapezius (LTR: Yes)	3 sessions, 1 per week: Shock waves in upper trapezius	↓ Pain ↓ NDI ↑ PPT
Gallego- Sendarrubias et al. (2020)	2 sessions with 1 week interval: DN trapezius and levator scapulae (LTR: Yes) + manual therapy	2 sessions with 1 week interval: sham DN + manual therapy	↓ Pain ↑ PPT ↑ ROM ↓ NDI
Stieven et al. (2020)	4–6 sessions in 4 weeks: DN upper trapezius, middle trapezius, multifidus, splenius, and levator scapulae (LTR: Yes) + manual therapy (cervical and thoracic mobilization) and exercise	4–6 sessions in 4 weeks: manual therapy (cervical and thoracic mobilization) and exercise	↓ Pain
Gattie et al. (2021)	7 sessions in 4 weeks: DN trapezius, levator scapulae, splenius capitis, semispinalis, spinalis capitis, multifidus, and suboccipital muscles (LTR: Yes) + manual therapy + exercise	7 sessions in 4 weeks: sham DN + manual therapy + exercise	There are no differences between DN and sham DN
Leon-Hernandez et al. (2021)	2 sessions, 1 per week: DN upper trapezius (LTR: Yes) + 15 min of percutaneous needle electrical stimulation (low frequency versus high frequency)		↓ Pain (There are no differences between DN modalities)
Stieven et al. (2021)	1 session unilateral DN upper trapezius (LTR: Yes)	1 session: myofascial release or sham DN	↑ PPT ↓ Pain
Valiente-Castrillo et al. (2021)	6 sessions in 2 weeks: DN upper trapezius, levator scapulae, splenius, and multifidus (LTR: Yes) + self-stretching + 3 sessions 30′ therapy education for one of the experimental groups	10 sessions in 2 weeks: 15 min TENS and 15 min Microwave + self-stretching	↓ Pain ↓ NDI ↓ Kinesiophobia ↓ Catastrophism ↓ Depression ↓ Anxiety ↓ Fear Pain ↑ Pain Attitudes

Abbreviations. DN: Dry Needling; LTR: Local Twitch Response; PPT: Pressure Pain Threshold; ROM: Range of Motion; NDI: Neck Disability Index.

## Data Availability

Not applicable.

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
