# Peer review of "Dry Needling in Physical Therapy Treatment of Chronic Neck Pain: Systematic Review"

_jcm, 2022, doi:10.3390/jcm11092370_

Round 1
Reviewer 1 Report
I would like to applaud the others for the work they have done with this study. I think more studies like this are needed where we look at common themes in the literature to help figure out what is best practice. I do feel this study will contribute to the literature, however, I suggest some changes. With all due respect I believe this study is actually a Scoping Review of the literature, and less of a systematic review. I would consider looking into this change since outcomes were not compared or assessed statistically.
Line 38: As part of the examination of the cervical spine, I suggest the authors speak to the importance of a comprehensive clinical examination of the cervical spine ( range of motion assessment, screening of joint mobility and neurological testing such as myotomes, dermatomes and reflexes) and reference this as they did for NPRS, VAS, NDI.
Line 51: The authors write “Dry needling is defined as an invasive physiotherapy technique used in the treat- 51 ment of neuromusculoskeletal disorders”. I would suggest that they label this a “minimally invasive technique”
Line 65: The authors write “local spasm reaction”. For consistency in the literature I would suggest calling it a local twitch response and describe briefly what it is. This would be a useful reference to include
Perreault, T., Dunning, J. and Butts, R., 2017. The local twitch response during trigger point dry needling: is it necessary for successful outcomes?. Journal of bodywork and movement therapies, 21(4), pp.940-947.
Line 67: consider finding another word for “hypertensive” in regard to describing trigger points. Perhaps just saying “ they are associated with motor abnormalities is enough here “
Line 81: Do the authors mean systematic review, because here they say “bibliographic review” and that is used multiple times throughout the text.
Line 99: It is more appropriate to have specific dates in regards to the search. It should be clearly written the Month and year the search ranges from. Ex DEC 2021 to JAN 2016.
Figure 1: line 21, putting multiple periods ……. At the end of a word is probably not appropriate for consistency in a flow diagram.
*Tables = there are no captions or keys in the tables to spell out in full the abbreviations, this must be corrected.
Lines 174-194: This part of the discussion should really be part of the results section of this review piece. Another table may showcase some of the important findings that you are discussing here as well. I would suggest reworking this into the results section and saving the discussion for the implications of the findings and what research supports thus far.
Lines 217-219: In Hongs technique the needle does not exit the tissue, it is retracted to the subcutaneous tissue and then redirected into another region of the trigger point. I would be careful in how this is described.
Author Response
Response to reviewer 1 Coments.
Point 1. With all due respect I believe this study is actually a Scoping Review of the literature, and less of a systematic review. I would consider looking into this change since outcomes were not compared or assessed statistically.
Response 1. Thank you very much for your careful review. It is possible to consider this article as a scoping review of the action guidelines in the treatment of chronic neck pain with dry needling. However, although this research does not include statistical analysis as a meta-analysis, it was carried out following the PRISMA guidelines.
Point 2. Line 38: As part of the examination of the cervical spine, I suggest the authors speak to the importance of a comprehensive clinical examination of the cervical spine (range of motion assessment, screening of joint mobility and neurological testing such as myotomes, dermatomes and reflexes) and reference this as they did for NPRS, VAS, NDI.
Response 2. Thank you very much for make better our manuscript. Following the recommendations, references on neurological testing are included.
Point 3. Line 51: The authors write “Dry needling is defined as an invasive physiotherapy technique used in the treat- 51 ment of neuromusculoskeletal disorders”. I would suggest that they label this a “minimally invasive technique”
Response 3. We are very thankful with the improvements proposed by the reviewer. The concept of “minimally invasive” is included in the new version.
Point 4. Line 65: The authors write “local spasm reaction”. For consistency in the literature I would suggest calling it a local twitch response and describe briefly what it is. This would be a useful reference to include
Perreault, T., Dunning, J. and Butts, R., 2017. The local twitch response during trigger point dry needling: is it necessary for successful outcomes?. Journal of bodywork and movement therapies, 21(4), pp.940-947.
Response 4. In the corrected version of the manuscript, we update the content regarding local twitch response and include the recommended reference. Thank you very much for your careful review.
Point 5. Line 67: consider finding another word for “hypertensive” in regard to describing trigger points. Perhaps just saying “they are associated with motor abnormalities is enough here “
Response 5. Thanks to the reviewer for this recommendation. Updates include definition of myofascial trigger points as “hypersensitive areas of muscle fibers associated with motor abnormalities”.
Point 6. Line 81: Do the authors mean systematic review, because here they say “bibliographic review” and that is used multiple times throughout the text.
Response 6. Thanks to the reviewer for this concern. Following the recommendations, the terminology is adapted throughout the text.
Point 7. Line 99: It is more appropriate to have specific dates in regards to the search. It should be clearly written the Month and year the search ranges from. Ex DEC 2021 to JAN 2016.
Response 7. We are very thankful with the improvements proposed by the reviewer the search was conducted between October and November 2021.
Point 8. Figure 1: line 21, putting multiple periods ……. At the end of a word is probably not appropriate for consistency in a flow diagram.
Response 8. We are so grateful with your appreciation. The flow diagram is modified by removing the points (…) and indicating “Records excluded: Systematic reviews, case reports and study projects”
Point 9. *Tables = there are no captions or keys in the tables to spell out in full the abbreviations, this must be corrected.
Response 9. Thank you very much for make better our manuscript. Corrections to tables include explanation of abbreviations.
Point 10. Lines 174-194: This part of the discussion should really be part of the results section of this review piece. Another table may showcase some of the important findings that you are discussing here as well. I would suggest reworking this into the results section and saving the discussion for the implications of the findings and what research supports thus far.
Response 10. Thanks to the comments of the reviewers this section is corrected. The information of the indicated paragraph is transferred to the results section. We are very thankful with the improvements proposed by the reviewer.
Point 11. Lines 217-219: In Hongs technique the needle does not exit the tissue, it is retracted to the subcutaneous tissue and then redirected into another region of the trigger point. I would be careful in how this is described.
Response 11. The explanation regarding the Hong’s technique is included “in which the needle is retracted into the subcutaneous tissue and then redirected to another region of the trigger point without leaving the tissue”. We are so grateful with your appreciation.

Reviewer 2 Report
This is a review of dry needling for people with chronic neck pain.
Introduction
- Line 21. The definition lacks a reference.
- Line 27. Please add the new classification of chronic pain by the World Health Organization. Explain if chronic neck pain included in this review was primary or secondary chronic pain.
- Line 51. Dry needling is not exclusive of physiotherapy. Please explain that in traditional Chinese medicine there is a technique called Ah-shi needling, which is dry needling of the most painful spot in the muscle.
- Line 62. Is superficial dry needling a placebo intervention?
- Line 64. Please explain what is myofascial trigger points. This is the first time it is mentioned in the manuscript, and it requires some description.
- Line 66-67. This line suggests that the authors of this review already have a pre-conceived idea of the effectiveness of dry needling “Thus, dry needling is a very effective treatment option for myofascial trigger points 66 (hypersensitive and hypertensive areas of muscle fibers) [23].
- Line 75. It is unclear what are the goals of this review. Is it to “deepen knowledge” or to do a systematic evaluation of the “effectiveness of dry needling”?
Methods.
- Line 81. Change this sentence. It is not a bibliographic review. It is a systematic review (as the title suggests).
- Line 81-82. Please articulate the objectives very clearly here.
- Please add the definitions used for inclusion/exclusion criteria. What was the definition of chronic pain? What happened if the trial included people with mixed acute and chronic pain? What happened if the study included people with mixed low back and neck pain? How was neck pain diagnosed, did they have whiplash? Did they have radiculopathy? Did you exclude studies that were not done by physiotherapists? Did you exclude studies that also added traditional Chinese medicine acupuncture?
- Was this review limited to randomized clinical trials? The word randomized is not mentioned. If this review was opened to non-randomized clinical trials, please explain why was that allowed? And how these studies were combined in the analyses. Were randomized trials given more weight than non-randomized studies? Were their quality assessed differently?
- Line 83 and 99. A systematic review of effectiveness that includes only trials conducted in the previous 5 years is very flawed.
- Line 97. A search strategy that only contains two terms is flawed. I see here a lot of terms that are missing to find studies for this review:
neck/ or neck muscles/ or exp cervical plexus/ or exp cervical vertebrae/ or spinal nerve roots/
exp arthritis/ or exp myofascial pain syndromes/ or fibromyalgia/ or spondylitis/ or exp spinal osteophytosis/ or spondylolisthesis/
whiplash injuries/ or cervical rib syndrome/ or torticollis/ or cervico‐brachial neuralgia.ti,ab,sh. or exp radiculitis/ or polyradiculitis/ or polyradiculoneuritis/ or thoracic outlet syndrome/
- Line 92. How were these outcomes included /excluded? For example, if quality of life was measured with a non-valid instrument, would you include this outcome in the review?
- Line 104. What are the items included in the PEDro scale? Is this a valid measure of trial quality? Please add references. Please add the questions of the PEDRo scale in an appendix, and the possible answers.
- Methods: missing information about who and how many assessors applied the inclusion/exclusion criteria, and quality assessment, and data extraction.
- Methods is missing how the authors decided to analyze the results. How they decided to make a qualitative or quantitative analysis?
- Line 139. How many were randomized and non-randomized?
- Table 1. Instead of showing the PEDro scale (should be PEDro score, not scale), please show the answers to each of the 10 questions. Maybe you need to show this in an appendix.
- Table 1. Add the study design. Randomized or not randomized.
- Table 1. Add the country where the studies were conducted.
- Table 1. Add the settings: was it done by physiotherapists? Was it outpatient or inpatient? Was it in a hospital, community, private clinic, or a pain clinic?
- Table 2. It is not enough just to show the direction of effect. Need to show the original data with means, standard deviations, or relative risks with SDs. Maybe you need an appendix to show all these data.
- Line 178. The first time that the word “placebo” appears is in the Discussion. This is very strange. I would have expected to see data in the results, both table and text. How many trials compared to a placebo? Was the placebo acceptable? Maybe there is an opportunity to combine some trials in a meta-analysis. Why did the authors do not combine them?
- Line 180. It is the first time that the authors talk about the study designs “all studies were randomized clinical trials”. This information should be given before. In the methods, did the authors only select randomized trials? In the results, this should be explained.
- Was the method of randomization appropriate, was concealment of allocation done well?
Discussion
- A lot of this information here should be reported in the results. More details should be added to the methods.
Conclusions
- I think that the conclusion is not convincing. The authors do not show the results of the individual trials, the quality is uncertain and the search strategy was limited to 5 years.
Author Response
Response To Reviewer 2 Coments
Point 1. Line 21. The definition lacks a reference.
Response 1. We are so grateful with your appreciation. The reference was pointed at the end of the second sentence, this error has been corrected in the new version.
Point 2. Line 27. Please add the new classification of chronic pain by the World Health Organization. Explain if chronic neck pain included in this review was primary or secondary chronic pain.
Response 2. We are very thankful with the improvements proposed by the reviewer. Updated version included the information recommended.
Point 3. Line 51. Dry needling is not exclusive of physiotherapy. Please explain that in traditional Chinese medicine there is a technique called Ah-shi needling, which is dry needling of the most painful spot in the muscle.
Response 3. Thank you very much for your careful review. This paragraph has been modified. We add more information about the traditional Chinese medicine acupuncture.
Point 4. Line 62. Is superficial dry needling a placebo intervention?
Response 4. Although in superficial dry needling the needle does not go deep into the muscle, the sources consulted indicate that it is a treatment option that can provide analgesia due to the mechanical stimulus generated. Thank you very much for make better our manuscript.
Point 5. Line 64. Please explain what is myofascial trigger points. This is the first time it is mentioned in the manuscript, and it requires some description.
Response 5. Thanks to the reviewer for this recommendation. Updates include definition of myofascial trigger points as “hypersensitive areas of muscle fibers associated with motor abnormalities”.
Point 6. Line 66-67. This line suggests that the authors of this review already have a pre-conceived idea of the effectiveness of dry needling “Thus, dry needling is a very effective treatment option for myofascial trigger points 66 (hypersensitive and hypertensive areas of muscle fibers) [23].
Response 6. The available resources suggest that the use of dry needling could be considered within the treatment of trigger points. The text is modified to clarify the position of the authors “dry needling could be a treatment option”. Thank you very much for your careful review
Point 7. Line 75. It is unclear what are the goals of this review. Is it to “deepen knowledge” or to do a systematic evaluation of the “effectiveness of dry needling”?
Response 7. We are so grateful with your appreciation. Updated manuscript includes “therefore, the main objective is to do a systematic evaluation of the effectiveness of dry needling in treatment of chronic neck pain”.
Point 8. Line 81. Change this sentence. It is not a bibliographic review. It is a systematic review (as the title suggests).
Response 8. Thanks to the reviewer for this concern. Following the recommendations, the terminology is adapted throughout the text.
Point 9. Line 81-82. Please articulate the objectives very clearly here.
Response 9. Thanks to the reviewer for this recommendation. The text is modified including the objective clearly in the recommended site.
Point 10. Please add the definitions used for inclusion/exclusion criteria. What was the definition of chronic pain? What happened if the trial included people with mixed acute and chronic pain? What happened if the study included people with mixed low back and neck pain? How was neck pain diagnosed, did they have whiplash? Did they have radiculopathy? Did you exclude studies that were not done by physiotherapists? Did you exclude studies that also added traditional Chinese medicine acupuncture?
Response 10. Thank you very much for your careful review. Inclusion/exclusion criteria are defined in update manuscript following the recommendations. Chronic pain definition is updated based on the new classification of chronic pain by the World Health Organization. Trials includes chronic patients with neck pain only. The studies included were done by physiotherapist. Studies based on acupuncture were excluded.
Point 11. Was this review limited to randomized clinical trials? The word randomized is not mentioned. If this review was opened to non-randomized clinical trials, please explain why was that allowed? And how these studies were combined in the analyses. Were randomized trials given more weight than non-randomized studies? Were their quality assessed differently?
Response 11. We are very thankful with the improvements proposed by the reviewer. The review was limited to randomized clinical trials. Updated version of the manuscript indicates the word randomized to describe the clinical trials included. All included articles were randomized clinical trials.
Point 12. Line 83 and 99. A systematic review of effectiveness that includes only trials conducted in the previous 5 years is very flawed.
Response 12. Thanks to the reviewer for this concern. The limitation to the last five years is proposed in order to know and analyze the most recent and updated references regarding invasive physiotherapy treatment, due to the growth of dry needling techniques in recent years.
Point 13. Line 97. A search strategy that only contains two terms is flawed. I see here a lot of terms that are missing to find studies for this review:
neck/ or neck muscles/ or exp cervical plexus/ or exp cervical vertebrae/ or spinal nerve roots/ exp arthritis/ or exp myofascial pain syndromes/ or fibromyalgia/ or spondylitis/ or exp spinal osteophytosis/ or spondylolisthesis/ whiplash injuries/ or cervical rib syndrome/ or torticollis/ or cervico‐brachial neuralgia.ti,ab,sh. or exp radiculitis/ or polyradiculitis/ or polyradiculoneuritis/ or thoracic outlet syndrome/
Response 13. We are so grateful with your appreciation. The search is focused on these terms in order to analyze the updated evidence easily ac-cessible to the health professional. It would be possible to include more combined terms, however, intending to show the results that the reader could quickly find. The research does not apply terms such as arthritis, fibromyalgia or whiplash, since the search focuses on chronic neck pain and primary origin, not associated with trauma or systemic cause.
Point 14. Line 92. How were these outcomes included /excluded? For example, if quality of life was measured with a non-valid instrument, would you include this outcome in the review?
Response 14. Thank you very much for your careful review. Those results that met the established criteria were included. The articles resulting from the search guidelines were analyzed in detail, the selection of articles included investigations that had valid measurement instruments.
Point 15. Line 104. What are the items included in the PEDro scale? Is this a valid measure of trial quality? Please add references. Please add the questions of the PEDRo scale in an appendix, and the possible answers.
Response 15. PEDro is Physiotherapy Evidence Database, this database is the main reference for finding out the most up-to-date evidence in physiotherapy and assessing the effects of interventions and treatments. PEDro scale includes 11 items that allow assessing the methodological quality of randomized clinical trials with a final score range of 0 to 10, the items included in the PEDro scale can be consulted in the appendix A in updated version. PEDro scale has excellent reliability for use in systematic reviews of randomized clinical trials (https://doi.org/10.1093/ptj/83.8.713). Questions of the PEDro Scale are available in appendix A. We are very thankful with the improvements proposed by the reviewer.
Point 16. Methods: missing information about who and how many assessors applied the inclusion/exclusion criteria, and quality assessment, and data extraction.
Response 16. Thank you very much for your careful review. The methodology information is corrected and expanded in the updated manuscript. Two independent reviewers performed the search and screened the articles, these reviewers applied inclusion/exclusion criteria, later, another reviewer supervised the systematic review and quality assessment, and data extraction.
Point 17. Methods is missing how the authors decided to analyze the results. How they decided to make a qualitative or quantitative analysis?
Response 17. We are so grateful with your appreciation. Authors decided to make a qualitative analysis due to heterogeneity in outcome measurement precluded statistical integration with guarantees.
Point 18. Line 139. How many were randomized and non-randomized?
Response 18. We are so grateful with your appreciation. All included articles were randomized clinical trials.
Point 19. Table 1. Instead of showing the PEDro scale (should be PEDro score, not scale), please show the answers to each of the 10 questions. Maybe you need to show this in an appendix.
Response 19. Thanks to the reviewer for this recommendation. Questions of the PEDro Scale and score details of randomized clinical trials selected are available in appendix A.
Point 20. Table 1. Add the study design. Randomized or not randomized.
Response 20. Thanks to the reviewer for this concern. Table did not include the study design because all included articles were randomized clinical trials.
Point 21. Table 1. Add the country where the studies were conducted.
Response 21. Thank you very much for make better our manuscript. The information regarding the country of each study is included in the new appendix B.
Point 22. Table 1. Add the settings: was it done by physiotherapists? Was it outpatient or inpatient? Was it in a hospital, community, private clinic, or a pain clinic?
Response 22. We are very thankful with the improvements proposed by the reviewer. The information regarding the characteristics of each study is included in the new appendix B.
Point 23. Table 2. It is not enough just to show the direction of effect. Need to show the original data with means, standard deviations, or relative risks with SDs. Maybe you need an appendix to show all these data.
Response 23. Thank you very much for your careful review. The representation of the direction of the effect is proposed as a simple way to quickly visualize the results of each article.
Point 24. Line 178. The first time that the word “placebo” appears is in the Discussion. This is very strange. I would have expected to see data in the results, both table and text. How many trials compared to a placebo? Was the placebo acceptable? Maybe there is an opportunity to combine some trials in a meta-analysis. Why did the authors do not combine them?
Response 24. Thanks to the reviewer for this concern. Three of the studies apply sham DN as a placebo treatment option, making it necessary to delve into the conditions of this intervention The interventions could suggest that the use of placebo could have a place within the treatments. The performance of a meta-analysis was limited by the heterogeneity in the measurement of the results.
Point 25. Line 180. It is the first time that the authors talk about the study designs “all studies were randomized clinical trials”. This information should be given before. In the methods, did the authors only select randomized trials? In the results, this should be explained.
Response 25. Updated version of manuscript includes more information about randomization, we explain that all results included were randomized clinical trials in methods and results. Thank you very much for make better our manuscript.
Point 26. Was the method of randomization appropriate, was concealment of allocation done well?
Response 26. Randomization minimizes selection bias and favors similarity between groups. All studies were randomized clinical trials. We are very thankful with the improvements proposed by the reviewer.
Point 27. A lot of this information here should be reported in the results. More details should be added to the methods.
Response 27. Thanks to the comments of the reviewers this section is corrected. The information of the indicated paragraph is transferred to the results section. We are very thankful with the improvements proposed by the reviewer. Following the recommendations, the information has been expanded in the methods section.
Point 28. I think that the conclusion is not convincing. The authors do not show the results of the individual trials, the quality is uncertain and the search strategy was limited to 5 years.
Response 28. Thank you very much for your careful review. Conclusions are based on the individual results of each study because heterogeneity in outcome measurement precluded statistical integration with guarantees. The limitation to the last five years is proposed in order to know and analyze the most recent and updated references regarding invasive physiotherapy treatment, due to the growth of dry needling techniques in recent years.

Round 2
Reviewer 1 Report
This is a nice study that contributes to the literature. One further suggestion would be to include in table 2 if the study sought to elicit or did elicit local twitch responses. This has been included in recent systematic reviews on the subject of DN.
Author Response
Response to reviewer 1 Comments (Round 2).
Point 1. This is a nice study that contributes to the literature. One further suggestion would be to include in table 2 if the study sought to elicit or did elicit local twitch responses. This has been included in recent systematic reviews on the subject of DN.
Response 1. Thank you very much for your careful review. Updated table 2 includes information about local twitch responses.

Reviewer 2 Report
Thank you for making changes to the manusript.
I still disagree that a systematic review that limits its search to the last 5 years contains the answer about the effectiveness of an intervention.
If the authors can convince that there is no important randomized trial published prior to 5 years, then the results of this current review might be valid. But if there is a trial published prior to 5 years, then the results of this review are invalid and biased.
Author Response
Response to Reviewer 2 Comments (Round 2).
Point 1. Thank you for making changes to the manusript. I still disagree that a systematic review that limits its search to the last 5 years contains the answer about the effectiveness of an intervention. If the authors can convince that there is no important randomized trial published prior to 5 years, then the results of this current review might be valid. But if there is a trial published prior to 5 years, then the results of this review are invalid and biased.
Response 1. We are so grateful with your appreciation. Following the recommendations, the limitation to the last 5 years is removed. We reviewed previous relevant articles and included important randomized clinical trials in the updated version of the review Thank you very much for make better our manuscript.
